# HECTOR: Hybrid Editable Compositional Object References for Video Generation

**Guofeng Zhang** [1 2]  **Angtian Wang** [2 †]  **Jacob Zhiyuan Fang** [2]  **Liming Jiang** [2]  **Haotian Yang** [2]  **Alan Yuille** [1]
**Chongyang Ma** [2]

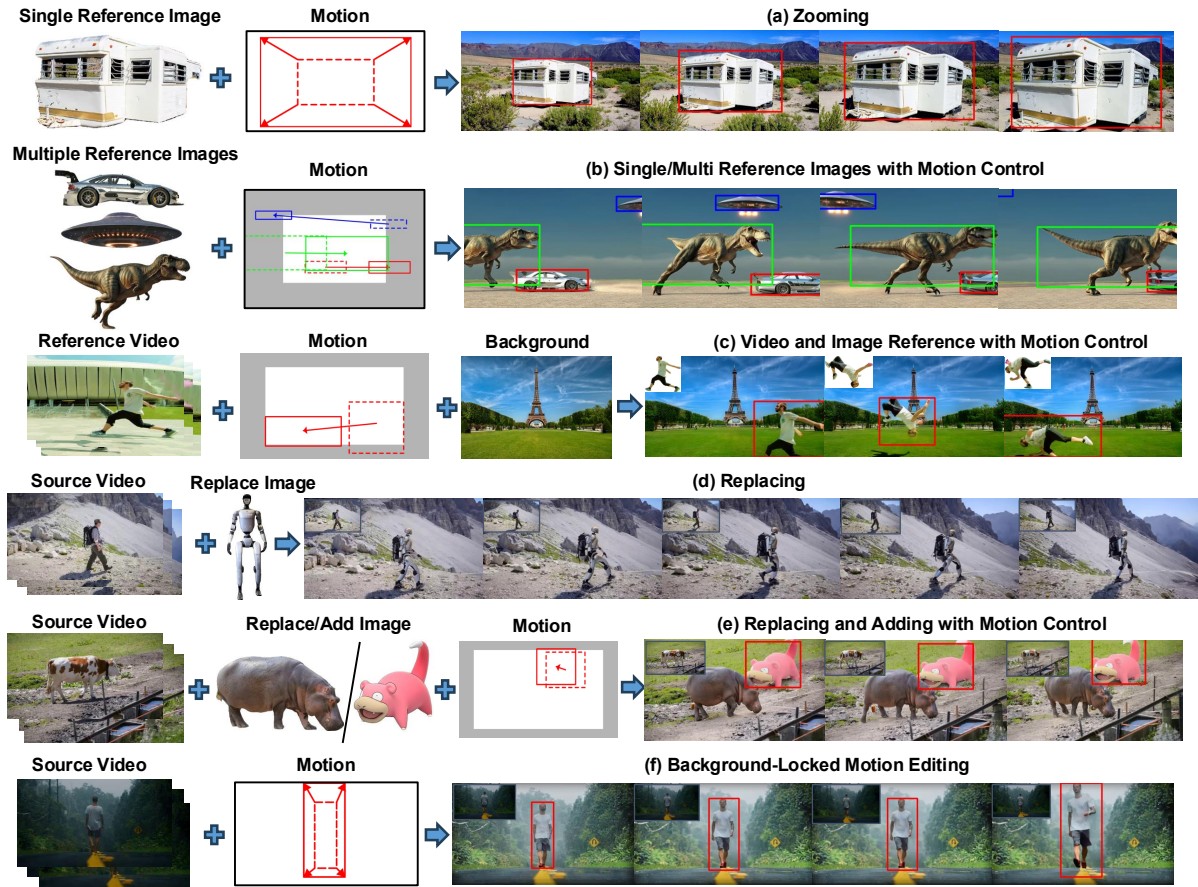

*Figure 1.* We propose HECTOR, a compositional, reference-guided video generation architecture. HECTOR supports conditioning on heterogeneous reference inputs (static images and/or dynamic videos) while enabling precise control over each referenced element's location, scale, and speed. Beyond that, HECTOR also accommodates diverse operations, including multi-object composition, camera-motion control (e.g., zoom-in/zoom-out), and reference-driven video editing such as object insertion, replacement as shown in the above.

## Abstract

Real-world videos naturally portray complex interactions among distinct physical objects, effectively forming dynamic compositions of visual elements. However, most current video generation models synthesize scenes holistically and therefore lack mechanisms for explicit compositional manipulation. To address this limitation, we propose HECTOR, a generative pipeline that enables fine-grained compositional control. In contrast to prior methods, HECTOR supports hybrid reference conditioning, allowing generation to be simultaneously guided by static images and/or dynamic videos. Moreover, users can explicitly

This work was done while Guofeng Zhang was an intern at ByteDance. †Angtian Wang is the project lead. [1]Johns Hopkins University [2]Bytedance, Intelligent Creation. Correspondence to: Guofeng Zhang <zhangguofeng1123@gmail.com>.

*Proceedings of the 43rd International Conference on Machine Learning*, Seoul, South Korea. PMLR 306, 2026. Copyright 2026 by the author(s).

specify the trajectory of each referenced element, precisely controlling its location, scale, and speed (see Figure 1). This design allows the model to synthesize coherent videos that satisfy complex spatiotemporal constraints while preserving high-fidelity adherence to references. Extensive experiments demonstrate that HECTOR achieves superior visual quality, stronger reference preservation, and improved motion controllability compared with existing approaches.

# 1. Introduction

The recent proliferation of diffusion-based generative models has revolutionized video synthesis, with Text-to-Video (T2V) (Liu et al., 2024b; Wang et al., 2023; Yang et al., 2024a; Wan et al., 2025) and Image-to-Video (I2V) (Blattmann et al., 2023; Zhang et al., 2023; Bar-Tal et al., 2024; Wan et al., 2025) paradigms enabling the creation of high-fidelity dynamic content for diverse applications, from entertainment to content creation. Despite these advancements, the practical utility in professional settings remains constrained by a lack of precise controllability. Standard approaches typically generate scenes holistically, where the user provides a high-level prompt but lacks the agency to dictate specific object behaviors or interactions.

To address this limitation, recent research has begun to explore fine-grained control mechanisms. Methods including Motion Prompting (Geng et al., 2025a), Tora (Zhang et al., 2025d), TGT (Zhang et al., 2025a), ATI (Wang et al., 2025b), and Wan-Move (Chu et al., 2025) have demonstrated the ability to guide motion through trajectories. However, these approaches often operate on video as a single entity. In this work, we push this exploration a step further by asking: *Can we generate video in a fundamentally compositional manner?* By decomposing a scene into distinct visual references, we aim to grant users explicit control over the appearance and motion of each individual component, including background, within the generated video.

Recent works have explored this along two main fronts. The first category focuses on instance-level customization, where methods such as DreamVideo (Wei et al., 2024a) and MotionBooth (Wu et al., 2024a) enable a single object to be localized via bounding box constraints. While effective for maintaining identity, these approaches typically rely on test time optimization for each specific reference, a process that is computationally expensive and difficult to scale to complex scenes with multiple interacting references.

A second line of research finetunes existing video generation models to integrate control signals. Notable examples include Tora2 (Zhang et al., 2025e), DreamVideo2 (Wei et al., 2024b) and VACE (Jiang et al., 2025). These method

integrate various conditional inputs in to existing models without requiring costly test-time optimization. While more efficient, these models face inherent difficulties in achieving true compositionality. In particular, they often exhibit degraded performance when processing multiple entities, frequently struggling to maintain precise boundaries or identity consistency as the scene complexity increases. Furthermore, these approaches lack explicit support for independent background conditioning and dynamic video references. While they can preserve identity from a static image, they are not designed to ingest video priors to maintain both the identity and the specific gestures of a subject.

To bridge this gap, we introduce Hybrid Editable Compositional Object References (HECTOR), a framework supporting both image and video-based references. To handle data, we adopt the Video Decompositor, which serves as both a curation and inference engine. Departing from rigid bounding box heuristics, it aggregates tracking points to derive precise motion paths and scales. This ensures superior temporal smoothness and stability, maintaining target consistency even during challenging occlusion events.

We adapt a DiT-based architecture to inject these signals via the Spatio-Temporal Alignment Module (STAM). STAM encodes diverse references into VAE latents, spatially organizing them to align strictly with trajectory-defined locations and timesteps. Specifically, it fuses image-based latents (for identity) and video-based latents (for gestures) into a unified conditioning tensor, gated by confidence masks and concatenated with the latent noise. This mechanism effectively binds visual identities and motion priors to precise spatial regions, enabling genuine compositional generation of coherent foreground and background entities.

Extensive experiments demonstrate that HECTOR generates video with strict identity and structural consistency. Uniquely, the framework supports dynamic object entry and exit without disrupting global temporal flow. Beyond generation, HECTOR unlocks powerful editing capabilities, including high-fidelity object replacement, addition, and background modification. By decoupling identity from motion, it further enables localized manipulations—such as altering an entity's speed or scale—while preserving the integrity of the surrounding scene (see Figure 1).

In summary, our contributions are as follows:

- We propose HECTOR, the first framework for fully compositional video generation. It enables precise, independent control over each element.

- We introduce the Spatio-Temporal Alignment Module (STAM), which processes both static and dynamic references spatially and temporally in the latent space.

- We present the Video Decompositor, a mechanism that

automatically extracts compositional structures from video data, supporting both accurate training data curation and flexible video editing during inference.

## Conflict of Interest Disclosure

Our model (HECTOR) was built utilizing some internal data provided by Bytedance, where co-authors are employed. But all evaluations are conducted on external, public datasets. No other conflict of interest exists.

## 2. Related Works

**Foundational video generation model.** Recent video generation advancements are driven by scaling diffusion transformers (DiTs), which enable high-fidelity, temporally coherent synthesis. Early latent-based approaches (Blattmann et al., 2023; Chen et al., 2024) democratized high-resolution generation, while leading open-source models (Wang et al., 2025a; Kong et al., 2024; Yang et al., 2024b) have established robust baselines for motion and realism. Recent works extend these foundations to complex tasks like long video generation (Liu et al., 2025; Zhang et al., 2025b; Henschel et al., 2025) and video editing (Cong et al., 2025; Guo et al., 2023). However, these frameworks typically generate scenes holistically, leaving fine-grained control and spatial compositionality largely underexplored.

**Reference-based video customization.** Reference-based models synthesize videos conditioned on specific subjects, primarily categorized into tuning-based (Wei et al., 2024a; Wu et al., 2025) and training-free (Yuan et al., 2025; Zhou et al., 2024) methods. Similarly, I2V models (Xing et al., 2024; Guo et al., 2024) and multi-concept frameworks (Huang et al., 2025; Deng et al., 2025a;b) focus on animating static reference images. However, these methods generally restrict users to static inputs and lack granular dynamic control. In contrast, our approach uniquely supports dynamic video references to integrate complex motion priors. By providing explicit guidance for reference motion and action, we bridge the gap between identity preservation and precise spatiotemporal controllability.

**Trajectory-controlled video generation.** Structure-based methods impose spatial layouts using bounding boxes or masks for precise alignment (Ma et al., 2024a;b; Hu & Xu, 2023), but these signals are rigid and labor-intensive. Zero-shot alternatives (Yu et al., 2024; 2023) reduce manual effort but often suffer from degraded controllability or quality (Su et al., 2023). Alternatively, sparse point-based trajectories offer intuitive, fine-grained control via 2D tracks (Wang et al., 2025b; Wu et al., 2024b; Geng et al., 2025b; Zhang et al., 2025c; Namekata et al., 2025; Shin et al., 2025; Wang et al., 2025c; Zhang et al., 2025a), yet often lack the flexi-

bility to compose complex scenes from mixed sources. We address this with a compositional pipeline integrating both static images and dynamic video references. By explicitly defining trajectory, scale, and speed, our approach ensures precise alignment while preserving reference fidelity.

## 3. Method

As illustrated in Figure 3, the HECTOR pipeline comprises two primary systems: the **Video Decompositor** (Section 3.2) and the **HECTOR Generative Model** (Section 3.3). The Video Decompositor is designed to decompose existing videos into distinct, manageable elements. This module serves a dual purpose: it processes training data for learning and extracts assets and layouts from videos during inference, which enables video editing capabilities. Built upon a pre-trained video diffusion backbone, the HECTOR model introduces a novel **Spatio-Temporal Alignment Module (STAM)**. This component is responsible for compositing individual elements back into a coherent video with precise spatiotemporal control. Finally, we detail our specialized training and inference setups in Section 3.4.

### 3.1. Preliminaries

**Diffusion Transformers (DiTs).** The generative process is defined within a latent manifold $\mathcal{Z}$, where a clean sample $\mathbf{z}_0$ is mapped to a Gaussian prior via a forward diffusion chain governed by a variance schedule $\{\beta_t\}_{t=1}^T$. The distribution of a noisy latent $\mathbf{z}_t$ at $t \in [1, T]$ is given by:

$$q(\mathbf{z}_t|\mathbf{z}_0) = \mathcal{N}(\mathbf{z}_t; \sqrt{\bar{\alpha}_t}\mathbf{z}_0, (1 - \bar{\alpha}_t)\mathbf{I}), \quad (1)$$

where $\bar{\alpha}_t$ represents the cumulative noise schedule. Modern architectures, including DiT, operate on a tokenized latent space. The latent $\mathbf{z}$ is partitioned into a sequence of spatio-temporal tokens $\mathbf{Z} \in \mathbb{R}^{N \times D}$, which are augmented with positional and temporal embeddings. The denoiser $\epsilon_\theta$ is optimized via the denoising objective:

$$\mathcal{L} = \mathbb{E}_{\mathbf{z}_0, \epsilon \sim \mathcal{N}(\mathbf{0}, \mathbf{I}), t, \mathcal{C}} \left[ |\epsilon - \epsilon_\theta(\mathbf{z}_t, t, \mathcal{C})|_2^2 \right]. \quad (2)$$

By modeling the denoising task as a sequence-to-sequence problem, DiTs demonstrate superior scalability and the ability to integrate diverse conditioning signals $\mathcal{C}$.

**Image-conditional video generation.** In image-conditioned settings, the generative process operates within a latent manifold $\mathcal{Z}$ established by a pre-trained VAE encoder $\mathcal{E}$. The transformer backbone processes a concatenated input tensor $\mathbf{X}_{in} = [\mathbf{z}_t, \mathbf{M}, \mathbf{z}_{cond}]$, where $\mathbf{z}_t$ is the noisy video latent and $\mathbf{M}$ is a multi-channel control mask. The structural prior $\mathbf{z}_{cond}$ is constructed by appending the VAE-encoded first frame with zero-filled latents for the subsequent frames. Within each transformer layer,

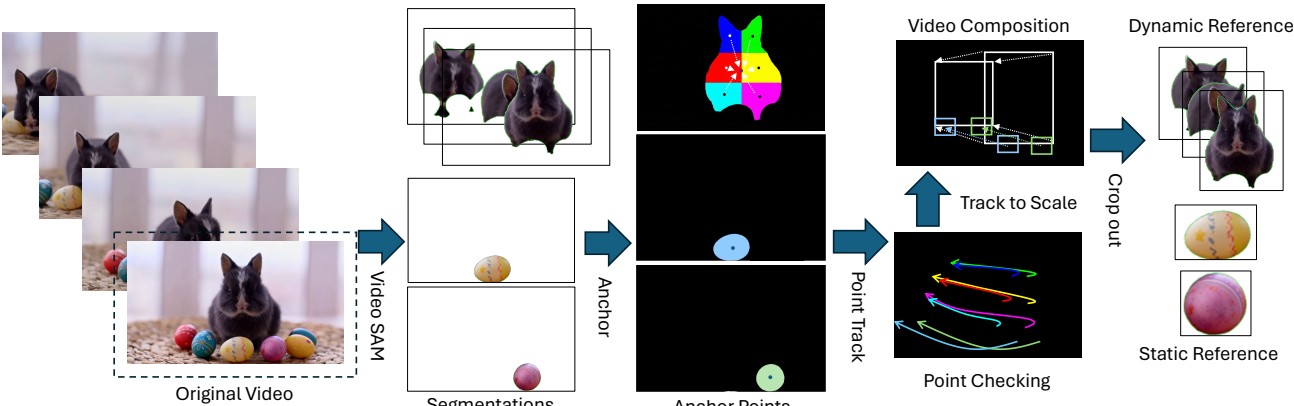

*Figure 2.* **Pipeline of the Video Decompositor**, which extracts video composition alongside dynamic and static references from a video. Specifically, Video SAM is first used to segment elements from the footage. Depending on the entity size, we place one or multiple anchor points on each object. A point tracking method is then used to propagate these selected anchors over time. We design a reference trajectory extraction method that converts the anchor tracks into a composition layout, capturing both the scale and translation of the entity. Finally, we crop each object from the original video using the computed spatial parameters to serve as the reference.

self-attention captures global spatio-temporal dependencies across video tokens $\mathbf{Z}$, while multi-head cross-attention integrates conditional information from $\mathcal{C}$ with $M$ heads. The output of the $m$-th head, $\mathbf{H}^{(m)}$, is defined as:

$$\mathbf{H}^{(m)} = \text{softmax}\left(\frac{\mathbf{Q}^{(m)}\mathbf{K}^{(m)\top}}{\sqrt{D_h}}\right)\mathbf{V}^{(m)}, \qquad (3)$$

where $D_h = D/M$. Queries $\mathbf{Q}$ are derived from video tokens $\mathbf{Z}$, while keys $\mathbf{K}$ and values $\mathbf{V}$ are projected from condition features $\mathcal{C}$. The final output concatenates all heads and applies a projection $\mathbf{W}^O$, giving $\left[\mathbf{H}^{(1)} \parallel \cdots \parallel \mathbf{H}^{(M)}\right]\mathbf{W}^O$. This configuration integrates high-dimensional spatial priors while preserving scalability, enabling HECTOR to finely compose hybrid references within the latent manifold.

**Trajectory-grounded motion modeling.** To provide a basis for controlled object dynamics, motion is formalized as a geometric path within the spatio-temporal volume. A trajectory $\mathcal{T}$ is defined as a sequence of $T$ time-indexed control anchors: $\mathcal{T} = \{\tau_t\}_{t=1}^{T}$, where $\tau_t = (\mathbf{p}_t, \mathbf{s}_t, v_t)$. Here, $\mathbf{p}_t \in [0,1]^2$ represents the normalized spatial coordinates of the object centroid, and $\mathbf{s}_t \in [0,1]^2$ denotes the relative spatial scale at frame $t$. To account for temporal boundaries such as object entry, exit, or occlusion, a binary visibility indicator $v_t \in \{0,1\}$ is incorporated to specify the presence of the object at each timestep. This representation allows for the modeling of continuous motion and scaling transitions, providing a structural precursor that grounds the generative process within the latent sequence $\mathbf{Z}$.

### 3.2. Video Decompositor

As shown in Figure 2, the Video Decompositor serves as the primary engine for constructing a robust trajectory-reference dataset, functioning both as a high-fidelity curation pipeline during training and a precise video-processing suite for editing-related inference. Here we introduce the modality of our Video Decompositor in details.

**Video captioning.** We utilize Qwen2.5-VL (Bai et al., 2025) to process the entire video, leveraging its multi-modal understanding to produce dense captions that comprehensively capture the overall scene, dynamics, and context.

**Object identification and anchor points sampling.** The Decompositor first segments objects within a reference frame $t_{ref}$ using SAM2 (Ravi et al., 2024) to establish precise pixel boundaries. Then, we adopt a patch partitioning strategy where the object's mask is dynamically divided into $K$ subregions based on its aspect ratio and pixel density. An anchor point is then sampled at the centroid of each patch. Also, we enforce a minimum patch size threshold, in which case, a single centroid anchor is sampled for small objects. Ultimately, this adaptive mechanism ensures we sample an spatially distributed set of anchor points for objects of varying shapes and sizes, providing a strong foundation for subsequent tracking and motion synthesis.

**Reference trajectory extraction.** Once anchor points are established, they are propagated across the temporal sequence using a point-based tracker, Cotracker3 (Karaev et al., 2024), to derive the trajectories as defined in preliminaries. To obtain the scale $\mathbf{s}_t$, we first calculate the base absolute scale $\mathbf{s}_{base} \in [0,1]^2$ of the object's bounding box at the reference frame $t_{ref}$, normalized by the image dimen-

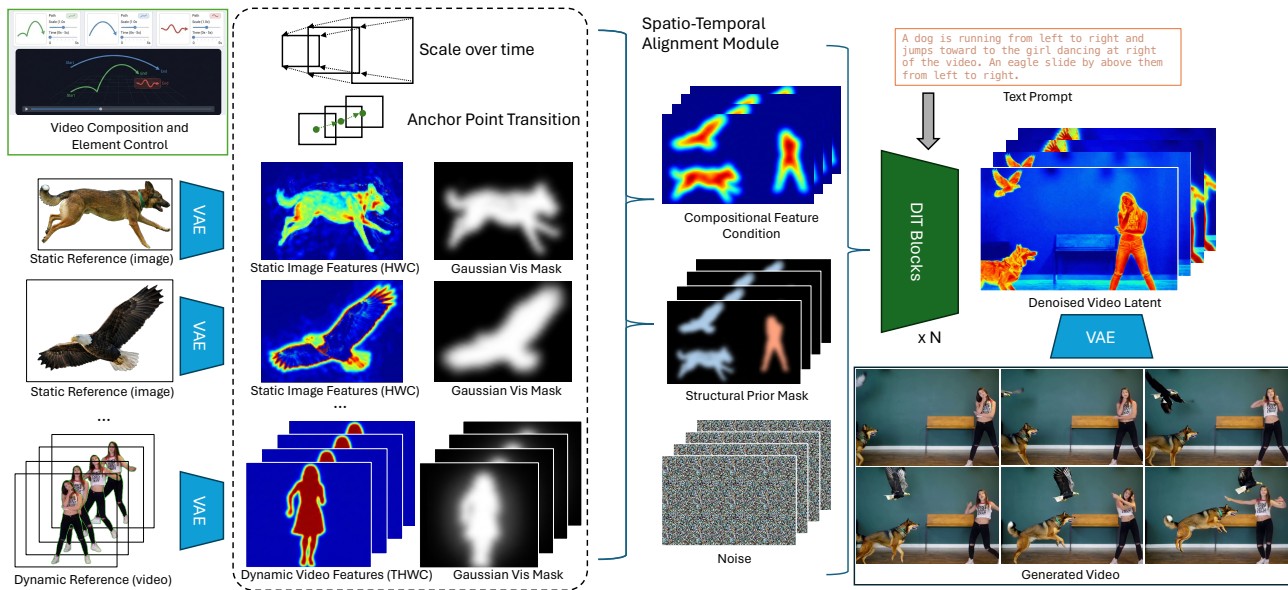

*Figure 3.* **Overview of the HECTOR framework**, which accepts hybrid inputs—static images and dynamic video references—alongside user-defined spatiotemporal layouts. The Spatio-Temporal Alignment Module (STAM) projects these references into the latent space using dynamic Gaussian masks to create aligned feature conditions. These conditions guide the DiT backbone to synthesize a unified video that preserves reference fidelity while strictly adhering to the specified motion trajectories.

sions $(W, H)$. We then compute the temporal scaling factor $\gamma_t$ by measuring the expansion or contraction of the point cluster $\{\mathbf{k}_{i,t}\}_{i=1}^{K}$ relative to the reference frame:

$$\gamma_t = \frac{1}{K} \sum_{i=1}^{K} \frac{|\mathbf{k}_{i,t} - \bar{\mathbf{k}}_t|_2}{|\mathbf{k}_{i,t_{ref}} - \bar{\mathbf{k}}_{t_{ref}}|_2 + \epsilon} \quad (4)$$

where $\bar{\mathbf{k}}_t$ is the cluster centroid and $\epsilon$ is a small constant for numerical stability. The final scale anchor $\mathbf{s}_t$ is defined as the product of the base absolute scale and the temporal scaling factor: $\mathbf{s}_t = \gamma_t \cdot \mathbf{s}_{base}$. This Point-to-Scale formulation ensures that the trajectory reflects the object's physical footprint within the normalized image plane. By grounding the scale in the internal variance of tracked keypoints, the Video Decompositor provides a smoother motion prior than traditional bounding box heuristics, which are often prone to jitter. Furthermore, the visibility indicator $v_t$ is determined by the aggregation of tracker's confidence scores on sampled anchor points, allowing for the precise signaling of object entry and exit events within the trajectory $\mathcal{T}$.

### 3.3. HECTOR

**Tokenization and backbone.** Our framework operates on a latent video volume $\mathbf{z}_t \in \mathbb{R}^{T \times H \times W \times C}$, which is first flattened and partitioned into a sequence of spatio-temporal tokens $\mathbf{Z} \in \mathbb{R}^{L \times D}$. Here, $L = \frac{T \times H \times W}{s^2}$ denotes the total token count for a patch size $s$, and $D$ represents the latent dimension. Each transformer block is composed of three primary elements: (i) spatio-temporal self-attention over $\mathbf{Z}$ to capture long-range dependencies, (ii) cross-attention to inject global semantic features, and (iii) Adaptive Layer Norm (AdaLN) for timestep-based modulation.

**Spatio-Temporal Alignment Module (STAM).** To enable precise structural control, we follow the image-conditioned paradigm where the input is augmented by concatenating the noisy latent $\mathbf{z}_t$ with a structural conditioning latent $\mathbf{z}_{cond}$ and its associated multi-channel mask $\mathbf{M}$. This setup allows the backbone to leverage localized appearance priors directly within the tokenized manifold. We propose the Spatio-Temporal Alignment Module (STAM) to integrate heterogeneous reference signals—ranging from static image exemplars to dynamic video sequences—with the extracted trajectories $\mathcal{T}$. STAM serves as a bridge that transforms these discrete references into the aligned conditioning latent $\mathbf{z}_{cond}$ and its corresponding mask $\mathbf{M}$.

The alignment process begins by encoding each reference $n$ into the pre-trained VAE latent space. To unify the processing of heterogeneous inputs, we apply modality-specific temporal transformations. Static image features $\mathbf{F}_i$ are broadcast across the temporal dimension $T$, while dynamic video features $\mathbf{F}_v$ are temporally resampled with interpolation to align with the target sequence. We then employ trajectory-guided inverse warping to "place" these features into the empty latent canvas. For each timestep $t$, a sampling grid $\mathcal{G}_{n,t}$ maps target coordinates back to the reference

source based on the tracked centroid $\mathbf{p}_{n,t}$ and scale $\mathbf{s}_{n,t}$:

$$\hat{\mathbf{F}}_n = \text{GridSample}(\mathbf{F}_n^{ext}, \mathcal{G}_n), \quad \mathcal{G}_{n,t}(\mathbf{u}) = \frac{\mathbf{u} - \mathbf{p}_{n,t}}{\mathbf{s}_{n,t} + \epsilon}. \quad (5)$$

We compute distinct latent volumes for image references, $\mathbf{V}_i$, and video references, $\mathbf{V}_v$, along with their respective Gaussian softened visibility masks $\mathbf{M}_i$ and $\mathbf{M}_v$. The final conditioning latent $\mathbf{z}_{cond}$ is the sum of these branches:

$$\mathbf{z}_{cond} = \mathbf{V}_i + \mathbf{V}_v = \sum_{j \in \mathcal{I}} \hat{\mathbf{F}}_j \odot \mathbf{M}_j + \sum_{k \in \mathcal{V}} \hat{\mathbf{F}}_k \odot \mathbf{M}_k, \quad (6)$$

where $\mathcal{I}$ is the set of reference images and $\mathcal{V}$ is the set of reference videos. The mask $\mathbf{M}$ is constructed as a 4-channel tensor to explicitly guide the DiT on the source of the structural prior. It concatenates the individual modality masks with their union: $\mathbf{M} = [\mathbf{M}_i, \mathbf{M}_v, \mathbf{M}_{union}, \mathbf{M}_{union}]$, where $\mathbf{M}_{union} = \text{Clamp}(\mathbf{M}_i + \mathbf{M}_v, 0, 1)$. This multi-channel design allows the transformer backbone to distinguish between static appearance constraints and dynamic motion priors during the generative process. Finally, the input to HECTOR is formed by the channel-wise concatenation of the noisy video latent, the guidance mask, and the structural condition, yielding a unified tensor $\mathbf{X}_{in} = [\mathbf{z}_t, \mathbf{M}, \mathbf{z}_{cond}]$.

### 3.4. Training and Inference

**Training objective.** We optimize the model parameters $\theta$ following a flow-matching objective with velocity prediction. Let $\mathbf{z}_1$ denote the ground-truth video latent encoded by the VAE, and $\mathbf{z}_0 \sim \mathcal{N}(\mathbf{0}, \mathbf{I})$ denote the source Gaussian noise. For a timestep $t \in [0, 1]$, we define the forward process as a linear interpolation $\mathbf{z}_t = t\mathbf{z}_1 + (1 - t)\mathbf{z}_0$. The model $v_\theta$ is trained to predict the flow velocity $\mathbf{v}_t = \frac{d}{dt}\mathbf{z}_t = \mathbf{z}_1 - \mathbf{z}_0$. The training objective is formulated as:

$$\mathcal{L}(\theta) = \mathbb{E}_{t, \mathbf{z}_0, \mathbf{z}_1} \left[ |\mathbf{v}_t - v_\theta(\mathbf{z}_t, t, \mathcal{C})|_2^2 \right], \quad (7)$$

where $\mathcal{C}$ represents the union of all conditioning signals, including the global text embeddings and the trajectory-aligned structural priors $(\mathbf{z}_{cond}, \mathbf{M})$ derived from STAM. By minimizing this objective, the model learns to straighten the generative trajectory from noise to data, facilitating compositional video synthesis following given trajectories.

**Dynamic modality prioritization.** During inference, the independent trajectories of static image references and dynamic video references may intersect, leading to spatial ambiguities where features from both modalities compete for the same latent tokens, particularly confusing for the model if one reference is used as background. To resolve this, we introduce a foreground-background gating mechanism. This control allows the user to explicitly designate a priority modality (e.g., forcing a static object to

remain in the foreground). We compute an inverse gate $\mathbf{G}_{inv} = 1 - \mathbf{M}_{fg}$ based on the foreground object's mask and apply it to the background modality's structural prior: $\mathbf{z}_{cond}^{bg} \leftarrow \mathbf{z}_{cond}^{bg} \odot \mathbf{G}_{inv}$. This operation effectively ensures clean occlusion boundaries and preventing feature bleeding or ghosting artifacts in complex, multi-object compositions.

## 4. Experiments

We detail the experimental setup, including dataset curation and baseline configurations, in Section 4.1. Section 4.2 presents a comprehensive quantitative and qualitative comparison demonstrating the superiority of our method. Finally, we validate the contribution of each proposed component through ablation studies in Section 4.3.

### 4.1. Experimental Setup

**Dataset.** We train on an internal corpus of 2.4 million high-resolution clips, curated from five million for motion magnitude and aesthetic quality scores. For evaluation, we use the DAVIS (Pont-Tuset et al., 2017) benchmark, employing SAM2 to propagate initial masks for dense video ground truth. From these high-quality annotations, we derive bounding boxes, static image references, and dynamic video references for baseline evaluations. On average, there are 3–4 distinct references per video, providing a challenging setup for multi-object compositional synthesis.

**Evaluation metrics.** We evaluate 8 metrics across three aspects: overall quality, subject fidelity, and motion control precision. For overall quality and consistency, we employ CLIP image-text similarity (CLIP-T) to measure semantic alignment and Temporal Consistency (T. Cons.) to assess the smoothness of the generated sequence. To evaluate subject fidelity, we utilize four metrics: CLIP image similarity (CLIP-I) and DINO image similarity (DINO-I) for global appearance, along with their region-based counterparts, Region CLIP-I (R-CLIP) and Region DINO-I (R-DINO). Finally, we assess motion control precision using Mean Intersection over Union (mIoU) and Centroid Distance (CD), which measures overlap and the normalized spatial deviation between the generated and target trajectories. We obtain the bounding boxes of the generated subjects by applying Grounded-DINO (Liu et al., 2024a) and SAM2, initialized with the ground-truth segmentation masks. Details of these metrics and be found in the Appendix A.

**Implementation details.** We implement our framework using the Wan2.1 I2V 14B (Wan et al., 2025) model as the backbone. We finetune the entire model on a cluster of 64 GPUs for 200K steps. The training utilizes the AdamW optimizer with $\beta_1 = 0.9$, $\beta_2 = 0.999$, and a weight decay of 0.01. We employ a constant learning rate of $1 \times 10^{-5}$

*Table 1.* Quantitative comparison for HECTOR vs baselines with image-based references. Best results in **bold**, second-best underlined.

| Method | R-CLIP ↑ | R-DINO ↑ | CLIP-I ↑ | DINO-I ↑ | CLIP-T ↑ | T-Cons ↑ | mIoU ↑ | CD ↓ |
|---|---|---|---|---|---|---|---|---|
| | | | *Single-Object* | | | | | |
| MotionBooth | 0.6277 | 0.2113 | 0.5622 | 0.3264 | 0.2855 | 0.9645 | 0.1822 | 0.2920 |
| VACE (bbox) | 0.6623 | 0.2602 | 0.5757 | 0.3357 | **0.3363** | 0.9909 | 0.2191 | 0.2740 |
| VACE (mask) | 0.6613 | 0.2497 | 0.5636 | 0.3048 | 0.3067 | 0.9913 | 0.2087 | 0.2821 |
| Ours | **0.6905** | **0.4277** | **0.6081** | **0.3735** | 0.3306 | **0.9914** | **0.3912** | **0.1130** |
| | | | *Multi-Object* | | | | | |
| VACE (bbox) | 0.6712 | 0.2479 | 0.5572 | 0.2683 | **0.3445** | 0.9908 | 0.1845 | 0.2633 |
| VACE (mask) | 0.6913 | 0.2540 | 0.5594 | 0.2747 | 0.3421 | 0.9911 | 0.2274 | 0.2937 |
| Ours | **0.6951** | **0.4347** | **0.5697** | **0.2939** | 0.3432 | **0.9915** | **0.4005** | **0.1670** |

*Table 2.* Ablation Study.

| Method | R-DINO ↑ | CLIP-T ↑ | CD ↓ |
|---|---|---|---|
| Bbox training | 0.4085 | 0.3392 | 0.192 |
| Img-Ref only | 0.4296 | 0.3427 | 0.171 |
| w/o Gaussian mask | 0.4211 | 0.3315 | 0.178 |
| **Ours** | **0.4347** | **0.3432** | **0.167** |

and apply gradient clipping at a threshold of 10.0. Video sequences are generated with a resolution of $832 \times 480$, spanning 81 frames at a frame rate of 16 fps.

**Baselines.** We evaluate our framework in two settings: Single-Object and Multi-Object. We compare against MotionBooth (Wu et al., 2024a) and VACE (Jiang et al., 2025). To benchmark structural control, we test VACE with bounding boxes (VACE-bbox) and trajectory-derived pseudomasks (VACE-mask). Notably, Tora2 (Zhang et al., 2025e) and DreamVideo2 (Wei et al., 2024b) are excluded due to lack of open source models. Quantitative score is restricted to the image-reference setting, as no baselines currently support both dynamic video referencing and explicit trajectory control. More baseline setup details are in the appendix B.

### 4.2. Experimental Results

**Quantitative comparison.** Table 1 presents quantitative results for single- and multi-object settings, where our method demonstrates superior performance across most metrics. Regarding overall consistency and quality, our approach achieves high temporal consistency (T-Cons) and competitive text-image alignment (CLIP-T). This indicates that the STAM module integrates precise structural control without compromising generative quality or video smoothness, effectively balancing semantic fidelity with dynamic motion. In terms of subject fidelity, our method consistently outperforms baselines by a significant margin on identity-preserving metrics, particularly R-DINO and DINO-I. This advantage persists in both single- and multi-object scenarios, confirming that our conditioning strategy preserves fine-grained appearance details far more effectively than standard bounding box or mask-based controls. Finally, our framework exhibits exceptional motion control precision, nearly doubling the accuracy of the strongest competitors in mIoU and Centroid Distance (CD). This confirms that our explicit trajectory alignment ensures strictly grounded motion, preventing the spatial drift often observed in baseline methods, even when coordinating multiple entities simultaneously.

**Qualitative results.** Fig. 4 compares our method with baselines using image-based references qualitatively. While MotionBooth and VACE show reasonable trajectory adherence in simple single-object scenarios, they struggle in complex conditions. MotionBooth often suffers from low fidelity. As shown, it fails to preserve the facial features and clothing of the man in the first row, causing identity drift. VACE maintains better fidelity in simple settings but exhibits weaker spatial control, degrading significantly in multi-object or overlapping scenes. In contrast, our method handles these complexities well, ensuring high-fidelity preservation and precise spatial alignment.

We further demonstrate the advantages of video-based references for precise editing in Fig. 5. Our framework effectively handles object replacement, seamlessly integrating reference objects onto moving subjects, and multi-subject generation, where distinct video references independently guide separate entities. Additionally, we illustrate "Background-Locked Motion Editing," enabling foreground manipulation while keeping the background strictly frozen to the original footage (see Fig. 1 for more).

### 4.3. Ablation Study

To validate the effectiveness of the core components in HECTOR, we conduct an ablation study under the multi-object

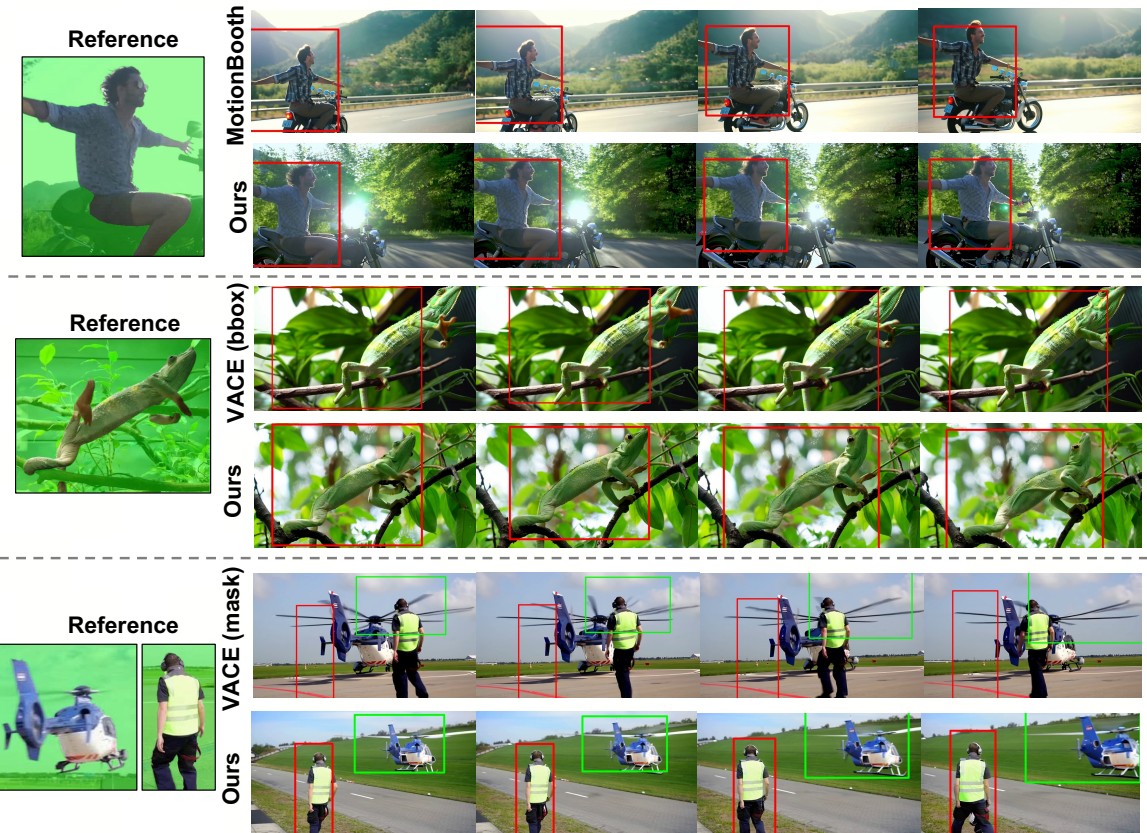

*Figure 4.* **Qualitative comparison against baselines.** We evaluate static reference-controlled video generation, as baselines are limited to this modality. The left column displays the source reference objects; for a fair experimental setup, we apply masks to crop the objects, ensuring all approaches receive only the object appearance without background context. The right columns show the resulting generated videos, illustrating the visual quality and precise spatial alignment with the input bounding box trajectories.

setting. We evaluate the impact of three key design choices: the trajectory-based scale in Video Decompositor, the image-video mixture of the training data, and the gaussian blur of mask condition. The results are summarized in Table 2.

**Trajectory-based scale.** We replace the anchor points tracking obtained scale with standard bounding box constraints, which leads to a notable degradation in both motion control (CD) and subject fidelity (R-DINO). This confirms that our point-based scale formulation provides significantly better structural guidance than bounding box constraints.

**Hybrid reference training.** Second, we investigate the influence of mixture of image and video training data by restricting the training pipeline to image references only. While this setting maintains competitive text alignment, the full model trained with hybrid video references achieves superior performance in motion adherence and visual quality.

**Gaussian masking.** Finally, we evaluate the impact of the soft conditioning mask by replacing the gaussian-softened masks with binary masks. The results show a decline in identity metrics, indicating that gaussian softening is essential

for blending reference features into the latent space.

## 4.4. Limitations and Failure Case Analysis

While HECTOR demonstrates robust compositional control, introducing fine-grained spatial and temporal constraints into a pre-trained diffusion model inherently presents certain boundary challenges:

**Upstream Tracking Dependencies:** The Video Decompositor relies on SAM2 and CoTracker3 for data extraction. In source videos with extreme motion blur or severe occlusion, tracking drift or incomplete segmentation can occur. Importantly, this does not compromise the global training process, as the DiT backbone is robust enough to handle occasional noisy trajectories in the training dataset. This dependency only limits specific video-editing tasks that require automatic trajectory extraction from source videos during inference. In these scenarios, any tracking errors can be easily mitigated through manual trajectory adjustments or by running the extraction process multiple times.

**Compositional Overlaps and Feature Blending:** Due to

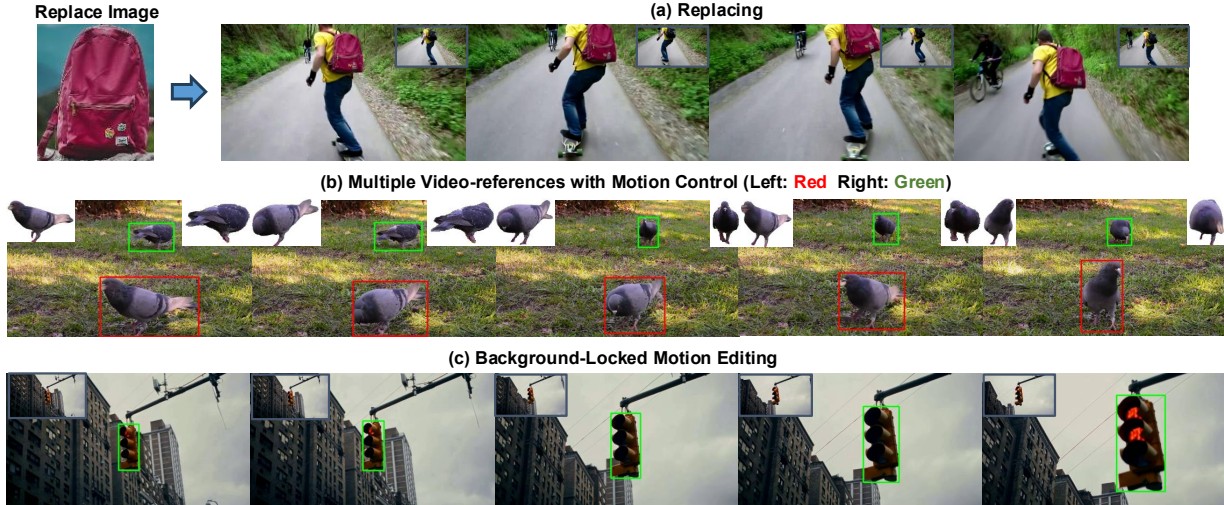

*Figure 5.* **Qualitative results for video reference.** We demonstrate our framework's versatility adopting video-based reference through three distinct applications: (a) Object Replacement, seamlessly transferring a reference object's identity onto a moving subject, (b) Compositional Multi-Subject Generation, where distinct video references independently control separate entities, and (c) Background-Locked Motion Editing, enabling precise foreground manipulations while keeping the background region frozen.

the additive nature of diffusion latents, when multiple foreground references follow intersecting trajectories without a predefined depth priority, feature competition can result in visual ghosting or boundary blending. We practically mitigate this using an explicit inverse gating mechanism to enforce user-defined depth priorities, though an end-to-end learned resolution for multi-object occlusion remains an open challenge in the field.

**Resolution of Small Entities:** When a referenced object occupies a minimal pixel area (e.g., thin accessories or small background elements), the compressed VAE latents inherently lack fine spatial detail. This can occasionally lead to lower texture fidelity or minor boundary artifacts along complex geometric edges, though the overall temporal tracking of the object remains stable.

## 5. Conclusion

We proposed HECTOR, a novel compositional video generation framework powered by the Video Decompositor and the Spatio-Temporal Alignment Module (STAM). Our experiments demonstrate that these designs are critical: replacing standard bounding boxes with our Decompositor's point-based tracking significantly reduced trajectory error. Furthermore, we observed that STAM's hybrid reference integration and Gaussian soft-masking were essential for achieving superior subject fidelity and identity preservation in complex multi-object scenes. These results confirm that explicit decomposition and alignment are key to bridging the gap between generative synthesis and precise video editing.

## Impact Statement

This paper presents advancements in controllable video generation, aiming to enhance tools for content creation, animation, and creative expression. We acknowledge that, like many generative models, our method could potentially be repurposed to generate misleading content. However, our focus on fine-grained trajectory control is primarily designed to improve the utility of generative AI for professional and artistic workflows. We will support the continued development of safeguards, such as deepfake detection and watermarking, to mitigate risks associated with synthetic media.

## Acknowledgement

This work was supported by the Office of Naval Research (ONR) under Grant N000142412696.

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

# A. Detailed Evaluation Metrics

To rigorously assess the performance of HECTOR, we evaluate eight distinct metrics across three primary dimensions: overall generative quality, subject fidelity, and motion control precision. This section provides the formal definitions and implementation procedures for each.

## A.1. Overall Quality and Temporal Consistency

These metrics measure the general aesthetic and semantic integrity of the generated videos, ensuring they align with the text prompt and remain stable over time.

- **CLIP Text Alignment (CLIP-T):** We utilize the pre-trained ViT-B/32 CLIP model to compute the cosine similarity between the global text prompt embedding $\mathbf{e}_{text}$ and the averaged visual embeddings of all generated frames $\{\mathbf{v}_t\}_{t=1}^{T}$.

- **Temporal Consistency (T-Cons):** To quantify the absence of flickering and the smoothness of the generated sequence, we calculate the average cosine similarity between CLIP image embeddings of consecutive frames:

$$\text{T-Cons} = \frac{1}{T-1} \sum_{t=1}^{T-1} \frac{\mathbf{v}_t \cdot \mathbf{v}_{t+1}}{\|\mathbf{v}_t\| \|\mathbf{v}_{t+1}\|} \tag{8}$$

## A.2. Subject Fidelity

Subject fidelity metrics assess how accurately the model preserves the identity and fine-grained details of the reference object. We employ both global and localized (region-based) metrics.

- **Global Fidelity (CLIP-I & DINO-I):** We measure the feature similarity between the reference object $n$ and the generated frames using CLIP and DINOv2. While CLIP captures high-level semantic identity, DINOv2 is utilized for its sensitivity to structural and granular textures.

- **Region-based Fidelity (R-CLIP & R-DINO):** To isolate the subject from background interference and evaluate local identity preservation, we compute the similarity between the reference image and the generated local subject. Specifically, we crop the generated frame using the ground-truth bounding box coordinate to extract the local region where the subject is intended to reside. By comparing these foreground crops directly against the original reference image, we provide a precise measure of identity fidelity that is independent of the synthesized background and overall scene composition. Plus, this can help to also evaluate the motion following performance of the generated videos.

## A.3. Motion Control Precision

To evaluate how strictly the model follows the spatial guidance provided by the user-defined trajectories, we calculate geometric alignment between the generated output and target bounding boxes.

- **Mean Intersection over Union (mIoU):** We extract the bounding boxes of the generated subjects $\mathbf{B}_{gen,t}$ and compute the overlap with the target boxes $\mathbf{B}_{target,t}$:

$$\text{mIoU} = \frac{1}{T} \sum_{t=1}^{T} \frac{\text{Area}(\mathbf{B}_{gen,t} \cap \mathbf{B}_{target,t})}{\text{Area}(\mathbf{B}_{gen,t} \cup \mathbf{B}_{target,t})} \tag{9}$$

- **Centroid Distance (CD):** We measure the Euclidean distance between the centroids of the generated and target boxes, normalized by the frame diagonal $D_{frame}$ to ensure scale invariance:

$$\text{CD} = \frac{1}{T} \sum_{t=1}^{T} \frac{\|\text{centroid}(\mathbf{B}_{gen,t}) - \text{centroid}(\mathbf{B}_{target,t})\|_2}{D_{frame}} \tag{10}$$

## A.4. Automated Ground Truth Extraction

To objectively obtain the bounding boxes $\mathbf{B}_{gen}$ from generated videos, we employ a multi-stage detection pipeline. We first use **Grounded-DINO** to identify candidate regions based on the subject's class name. These detections are then refined by **SAM2**, which is initialized with the ground-truth segmentation mask from the reference to ensure consistent tracking. The final bounding box is defined as the tightest axis-aligned rectangle enclosing the predicted SAM2 mask.

# B. Baseline Implementation Details

In this section, we provide the specific implementation protocols and configuration settings for the baselines used in our comparative study. All evaluations were conducted using the official open-source repositories and pre-trained weights to ensure reproducibility.

## B.1. MotionBooth (Wu et al., 2024a)

MotionBooth is a subject-driven video generation framework that utilizes a specialized training and inference pipeline to maintain subject identity.

- **Implementation:** We utilize the **finetuning-based version** of MotionBooth for our evaluation. To adapt the model to our task, we follow the subject-driven finetuning stage as described in the original paper, using the cropped reference object image as the primary appearance latent. This ensures the baseline has the highest possible chance of maintaining identity fidelity.

- **Trajectory Control:** MotionBooth's spatial control is strictly limited to **bounding box** inputs. During evaluation, we provide the same target bounding box trajectories used in HECTOR. However, unlike our point-based Decompositor, MotionBooth treats the bounding box as a region-based constraint, which can lead to less precise motion following for non-rectangular or fast-moving subjects.

- **Inference Settings:** We strictly adhere to the author's recommended configuration for high-fidelity subject preservation as specified in their public repository. Specifically, we employ the DDIM sampler with 50 inference steps and a guidance scale of 7.5. For the finetuning stage, we utilize the default learning rate and iteration count suggested for single-image subject injection.

## B.2. VACE (Jiang et al., 2025)

VACE is a generative model designed for versatile attribute-controlled video editing. We implement two variants to benchmark its structural control capabilities against our framework:

- **VACE-bbox:** For this implementation, the target bounding boxes are converted into the model's native layout-conditioning format. This configuration tests VACE's performance under sparse spatial constraints identical to the inputs received by HECTOR.

- **VACE-mask:** Since VACE is optimized for dense mask inputs, we derive pseudo-masks by placing mask of reference object along the trajectories. This represents an idealized setup for the baseline, providing it with pixel-level area constraints throughout the temporal sequence.

# C. LLM Usage

We used large language models (LLMs) in two limited ways: (i) to help generate and refine example content such as candidate captions/local prompts for qualitative demonstrations, and (ii) to assist with wording, formatting, and editing during manuscript preparation. All model-suggested text and prompts were reviewed, edited, or discarded by the authors; no experimental design, implementation, or quantitative analysis depended on LLM output.

