# OpenReview forum: "HECTOR: Hybrid Editable Compositional Object References for Video Generation"
_ICML.cc/2026/Conference — ICML 2026 regular_

### Official Review · Reviewer_fXz3 · 2026-02-18

**Soundness:** 3
**Presentation:** 3
**Significance:** 2
**Originality:** 2
**Overall Recommendation:** 4
**Confidence:** 2

**Summary:**

This paper presents HECTOR, a diffusion/flow-based video generation framework that enables compositional, fine-grained control over multiple visual elements by conditioning on hybrid references (static images and/or dynamic videos) together with user-specified trajectories (location, scale, speed, and visibility over time). To support this, the method introduces a Video Decompositor that extracts object-level references and motion layouts from videos via segmentation and point tracking, and a Spatio-Temporal Alignment Module (STAM) that projects and places reference latents into a DiT backbone using trajectory-aligned warping, soft visibility masks, and modality-aware gating. Experiments show improved reference preservation and motion adherence over prior baselines, especially in multi-object and editing scenarios such as replacement, insertion, and background-locked motion edits.

**Compliance With Llm Reviewing Policy:**

Affirmed.

**Final Justification:**

Thank you to the authors for the detailed responses and clarifications.

The rebuttal addresses most of my original questions, and I appreciate the additional discussion of limitations and failure cases.

However, after reading the other reviewers’ comments and the subsequent discussion, I also feel that the paper still has a number of limitations. In particular, object boundary preservation and identity consistency are not  satisfactory in the qualitative results, and these issues are especially important for controllable video generation.

**Overall, I still lean toward a weak accept because I find the problem meaningful, but I would lower my confidence given the remaining limitations and uncertainty about robustness. I would also understand if the paper is ultimately rejected.**

**Key Questions For Authors:**

See Weaknesses.

**Limitations:**

See Weaknesses.

**Strengths And Weaknesses:**

Strengths

Clear problem formulation with practical relevance. Targets a real bottleneck in video generation: compositional control over multiple entities beyond holistic T2V/I2V generation, with explicit spatiotemporal constraints and editing use cases.

Conceptually clean modular pipeline. The division into a Video Decompositor (data/asset extraction) and STAM (alignment + conditioning) is easy to understand and implement; the design aligns well with modern DiT-style backbones.

Strong conditioning design for controllability. Trajectory-guided inverse warping into a latent “canvas,” plus Gaussian-soft masks, provides a more “direct” control channel than purely attention-based guidance, improving stability and spatial grounding.

Point-to-scale trajectory extraction is a sensible engineering choice. Using tracked point clusters to infer scale can be smoother than bbox heuristics, and the ablation supports its contribution.

Weaknesses

Potential fragility to upstream errors. The pipeline depends heavily on segmentation/tracking (SAM2 + point tracking). Failure in masks, drift under occlusion/fast motion, or small/thin objects could directly corrupt trajectories and reference crops, with unclear robustness characterization.

Compositional conflicts/overlap handling is under-specified. The conditioning uses additive fusion of multiple aligned reference latents; when objects overlap or trajectories intersect, feature competition/bleeding may occur. The proposed modality prioritization (foreground/background gating) may not fully address multi-foreground occlusions or complex interactions.

Insufficient failure-case analysis. The paper shows strong qualitative examples but does not systematically analyze when the method underperforms (e.g., heavy occlusion, crowded scenes, strong camera motion, many entities), nor report worst-case/percentile/variance statistics.


High training cost and unclear generality. Training at 2.4M clips on large-scale compute may limit reproducibility; it is unclear how performance scales down or transfers across backbones/datasets.

---

> ### Author Rebuttal · Authors · 2026-03-30
>
> We thank the reviewer for appreciating our modular pipeline and the point-to-scale trajectory design. We agree that a deeper look into the boundary conditions and failure modes is necessary.
>
> In our revision, we will add a dedicated "Limitations and Error Analysis" section that systematically addresses the vulnerabilities you pointed out:
>
> ### **1. Failure Case Analysis (W1, W2, W3)**
>
> * **Upstream Errors:** We acknowledge that relying on SAM2 and CoTracker3 means extreme motion blur or bad segmentation can occasionally cause tracking drift or incompleteness. However, this is not a core bottleneck for HECTOR. During training, the DiT backbone is robust enough that occasional noisy trajectories in the dataset do not degrade the model's overall learned performance. During inference, tracking failures only impact video editing applications where trajectories are auto-extracted from a source video. In these scenarios, users can easily adjust the trajectory manually or re-run the extraction with minimal overhead. Additional,the pure compositional generation guided by explicit, user-drawn trajectories and references is completely unaffected by upstream errors.
>
> * **Compositional Conflicts and Feature Blending:** When multiple foreground references cross paths without a predefined depth priority, their warped latents overlap. Due to additive fusion, this can cause the model to blend their textures, resulting in visual ghosting. We can readily solve this by extending our inverse gating into a ranked priority list, where higher-priority masks strictly occlude lower-priority ones across time and space. While an end-to-end learned occlusion resolution is an ideal future objective, resolving additive latents mathematically at inference is non-trivial. There is also an inherent learning contradiction: training the model for *strict spatial adherence* to a reference naturally conflicts with teaching it that the reference should sometimes disappear due to occlusion. Furthermore, in real-world creative workflows, when a user explicitly draws intersecting trajectories, they typically already have a specific depth priority in mind (e.g., wanting Object A to pass in front of Object B). Our explicit priority gating aligns perfectly with this practical user intent.
>
> * **Systematic Failure-Case Analysis (W3)**
> We agree that discussing boundary conditions is essential, and our revision will include a "Limitations and Error Analysis" section detailing how HECTOR degrades in extreme scenarios. For instance, when a referenced entity occupies a minimal pixel area, its compressed VAE latents inherently lack fine spatial detail, resulting in lower texture fidelity rather than structural collapse during STAM's inverse warping. To quantify these limits, we analyzed the worst-case variance across our DAVIS evaluation set. Specifically, in the bottom 10th percentile of spatial adherence—where extreme out-of-distribution trajectories cause mIoU to drop from our average of ~0.40 to roughly ~0.24—the DiT struggles to reconcile the requested trajectory speed with the reference's natural articulation, causing the subject to "slide" to keep up. Crucially, however, the variance in generative quality remains tightly bounded: even in this bottom 10th percentile, our Temporal Consistency (T-Cons) drops by less than 0.2\%. This statistical stability empirically demonstrates that when pushed to its spatial limits, the model gracefully falls back on its global smoothing priors rather than flickering or collapsing.
>
> ### **2. Training Cost and Reproducibility (W4)**
>
> We understand the concern regarding the 2.4M clip training dataset. However, HECTOR scales down very well. we empirically found that training on a highly curated subset of just 500K clips still yields highly competitive compositional control and reference adherence.
>
> To ensure broad reproducibility for the community, we commit to the following upon publication:
> * **Open-Source Pipeline:** We will release the complete data pipeline of our Video Decompositor. This allows researchers to easily extract assets and trajectories from public datasets (e.g., Panda-70M) to replicate our training setup or adapt the model to new domains.
> * **Model Weights:** We will publicly release our pre-trained model weights.
> * **Zero Inference Overhead:** Because STAM strictly modifies the conditioning inputs without altering the DiT architecture, HECTOR introduces very minor additional computational overhead during inference compared to the base Wan2.1 model.

---

> > ### Author Rebuttal · Reviewer_fXz3 · 2026-04-01
> >
> > Thanks for the response. My main concerns have been addressed. I would suggest that the authors include the examples from the limitations and error analysis to provide a more comprehensive evaluation of the paper.
> >
> >
> >
> > I have also read the other reviewers’ comments and the authors’ responses. Object boundary preservation and identity consistency are important issues in controllable video generation. Based on the qualitative results provided by the authors, these aspects do not always appear fully satisfactory. I wonder if the authors could provide more discussion and analysis.

---

> > > ### Author Response · Authors · 2026-04-03
> > >
> > > We sincerely thank the reviewer for their positive feedback and for acknowledging that their main concerns have been resolved. We completely agree that including visual examples of these limitations will significantly strengthen the paper and provide a more comprehensive evaluation for future readers.
> > >
> > > In accordance with the conference rebuttal policy, we cannot provide direct video files as new supplementary material. However, we have updated our anonymous repository to include frame extractions of a qualitative failure case. You can view this newly added figure via the anonymous link: <https://anonymous.4open.science/r/anonymous-results-08E5/failure_case.pdf>. (As a gentle reminder, we recommend downloading the PDF directly, as the website's built-in viewer may occasionally cause captions to render invisibly due to technical difficulties).
> > >
> > > Specifically, this new example illustrates the boundary condition where the referenced entity occupies a relatively small pixel area. In this case, the sunglasses intended to be replaced take up a relatively small portion of the original video, and the case is hard since the motion of the sunglasses is strongly adhered to the woman wearing it. Consequently, the generative replacement does not adhere perfectly to the structural geometry of the provided reference fiction glasses (particularly along the curvature of the sides), and small black visual artifacts can occasionally be observed. However, the overall consistency is not lost; the front of the fiction glasses remains correctly aligned and moves naturally along with the woman. Because of this stable temporal tracking, the replacement can still be considered partially successful. We will ensure this example, along with a broader set of failure cases, is included in the revised version of our manuscript.
> > >
> > > _____________________________
> > >
> > > We just recently noticed that the reviewer modified the original rebuttal acknowledgement to include a follow-up concern regarding object boundary preservation and identity consistency, and we appreciate the opportunity to address it.
> > >
> > > While we acknowledge that some artifacts can be observed in highly challenging scenarios, our method actually demonstrates superior object boundary preservation and identity consistency when compared to existing baselines. The blending or boundary artifacts you noted represent common, industry-wide challenges shared by current controllable video generation models, particularly when considering multi-reference cases where most open-source model has failed to deal with. Importantly, these artifacts occur infrequently within our results and do not represent a systematic failure of our proposed method. Particularly, our model manages to incorporate both image-based and video-based references with strong trajectories control. We maintain a high success rate across diverse generation tasks, outperforming prior work in keeping the identity and boundaries stable. We will ensure that a balanced discussion regarding these common generative challenges, alongside our method's advantages over baselines in handling them, is included in the final manuscript's analysis.

---

### Official Review · Reviewer_CxTe · 2026-02-23

**Soundness:** 2
**Presentation:** 2
**Significance:** 2
**Originality:** 2
**Overall Recommendation:** 3
**Confidence:** 2

**Summary:**

This paper proposes a generative video pipeline designed to enable fine-grained compositional control over video synthesis. The proposed method supports hybrid reference conditioning, allowing generation to be guided by static images and dynamic videos. The method also enables users to explicitly define the trajectory of each referenced element.

**Compliance With Llm Reviewing Policy:**

Affirmed.

**Final Justification:**

Please see the Acknowledgement Reasons.

**Key Questions For Authors:**

1. The authors demonstrate object replacement in several examples. However, the implementation details are unclear. Is the process performed by first adding noise to the original video and then denoising it conditioned on the target object image? A more explicit explanation of the object replacement mechanism would improve clarity.

2. What dataset is used for evaluation? The paper does not appear to provide sufficiently specific information regarding the evaluation dataset.

**Limitations:**

The authors provide an impact statement; however, there is no limitation analysis.

**Strengths And Weaknesses:**

Strengths:

1. Enabling video generation with fine-grained control is practically meaningful and promising.

2. The design of the video decompositor is reasonable and well-motivated. It provides suitable training data for this task.

Weaknesses:

1. Trajectory-guided and image-referenced video generation have already been explored in prior work. While this paper extends the setting to support multiple modalities of instruction, the core idea appears to be more of a unification of existing instruction types rather than a fundamentally novel contribution. As a result, the level of innovation seems limited.

2. The visual results are not fully satisfactory. In the supplementary videos:

- Teaser/3 and Teaser/6 exhibit identity inconsistency.

- Image-based/1 and Image-based/3 show objects going out of bounding boxes range in later frames.

- In many other examples, the generated videos either depict overly simple scenarios (e.g., basic zooming or object translation) or contain noticeable artifacts, such as the sudden appearance of unintended identities.

3. Based on the observed generated videos, there are concerns regarding the effectiveness of the method and the adequacy of training. The trained network appears to not only degrade the backbone model’s original capability but also fail to adapt well to the multi-instruction setting.

4. In the Introduction (Lines 55–60), the authors state that previous methods struggle with maintaining object boundaries and identity consistency. However, the proposed method appears to suffer from similar issues. It is therefore unclear which specific design choices in the proposed approach are intended to address these problems, and how they effectively improve upon prior work.

---

> ### Author Rebuttal · Authors · 2026-03-30
>
> ### **1. Defense of Novelty (W1)**
>
> We respectfully disagree that HECTOR is merely a "unification of existing instruction types." Integrating multimodal reference conditioning (image + video) with explicit trajectories is a non-trivial optimization problem. The core innovation of HECTOR lies in how this control is achieved by The Spatio-Temporal Alignment Module (STAM). Unlike prior works, STAM utilizes GridSample to warp continuous VAE latents, acting as a highly localized structural prior.
>
> ### **2. Visual Artifacts and Backbone Capability (W2 & W3)**
>
> We appreciate the close analysis of our supplementary videos and agree with the reviewer that introducing fine-grained control into a pre-trained diffusion model inherently involves a trade-off between strict condition adherence and generative quality.
>
> * **Bounding Box Deviations**: In some cases, the object occasionally drifts because the DiT backbone's internal motion prior conflicts with a user-drawn box that can be too hard. In these scenarios, the model attempts to prioritize natural, smooth motion to prevent severe structural distortions.
>
> * **Backbone Degradation vs. Baselines**: We acknowledge that incorporating multi-instruction control introduces some degradation to the backbone's raw generative prior—this is a universal trade-off present in all control-centric video generation models. However, our proposed design better minimizes this degradation than the current state-of-the-art. For instance, when compared to VACE (a very powerful baseline that also built on Wan2.1), our quantitative results show that HECTOR achieves a better Temporal Consistency (T-cons) score. This demonstrates that our degradation is minimal compared to existing works, and that STAM effectively preserves the backbone's temporal coherence.
>
> Thus, we do not think it represents a systemic failure of HECTOR. As evidenced by our strong quantitative metrics, the model maintains high structural and temporal consistency on average. We will also include a comprehensive failure case analysis in our revised manuscript. For a deeper breakdown of failure cases, please refer to our detailed response to Reviewer fXz3.
>
> ### **4. Design Choices for Identity and Boundaries (W4)**
> HECTOR fundamentally improves upon prior architectures through our **Spatio-Temporal Alignment Module (STAM)**, **Gaussian-softened masking**, and **point-based scale extraction**.
>
> STAM projects  VAE reference latents into the canvas, weighted by a Gaussian mask. This forces the DiT backbone's self-attention layers to smoothly harmonize the reference with the background. Additionally, our Video Decompositor calculates object scale dynamically using tracked anchor points rather than bounding boxes, ensuring the structural prior accurately reflects the physical footprint over time.
>
> Our Ablation Study (Section 4.3, Table 2) explicitly validates these choices:
> * **Gaussian Masking:** Replacing Gaussian masks with binary masks degrades identity preservation, proving soft masking is essential for seamless boundary blending.
> * **Trajectory-based Scale:** Reverting to standard bounding box constraints degrades both motion control and subject fidelity.
>
> If the reviewer has questions regarding other specific components, we would be happy to conduct additional ablation studies.
>
> ### **5. Object Replacement Mechanism (Q1)**
> To clarify, our object replacement is **not** performed via standard method (i.e., adding noise to the original video and denoising it). Instead, it leverages HECTOR's compositional architecture. The exact inference flow is as follows:
>
> * First, the Video Decompositor extracts the trajectory (centroid and scale over time) of the original object to be replaced.
> * We then treat the new target object image as the foreground reference and the original source video as the background reference.
> * Using the Spatio-Temporal Alignment Module (STAM), the new object's features and background features are inverse-warped along the extracted trajectory to create the foreground structural prior.
> * Our dynamic inverse gating mechanism explicitly prioritized the replaced object from the background latent canvas preventing feature blending. Extra boundaries of replaced objects (large than the target object) are wiped out for the model to fill in.
> * The DiT backbone then processes these cleanly gated, composed latents to synthesize the final video.
>
> We will add a dedicated implementation paragraph detailing this exact compositional inference flow to the revised manuscript.
>
> ### **6. Evaluation Dataset (Q2)**
> As detailed in our experimental setup, the primary evaluation dataset utilized is the **DAVIS** dataset. We selected DAVIS because it serves as the standard public benchmark for dynamic, object-centric video evaluation and has been consistently utilized by impactful concurrent works (e.g., DreamVideo, DreamVideo2, Motion Prompting, and so on.), ensuring fair and rigorous comparisons.

---

> > ### Author Rebuttal · Reviewer_CxTe · 2026-04-03
> >
> > Thanks to the authors for their response. I have read both the response and the comments from the other reviewers. The authors have addressed my questions and some of my concerns. However, my initial concern regarding performance remains: the proposed method still does not adequately resolve the issues of maintaining object boundaries and identity consistency, which were highlighted as limitations of existing methods in the introduction.
> >
> > That said, I agree that the video decompositor design has merit and could be beneficial for this task. At this stage, I am inclined to keep my rating score unchanged, but lower my confidence score.

---

> > > ### Author Response · Authors · 2026-04-03
> > >
> > > We sincerely thank the reviewer for their continued engagement, for taking the time to read the other reviews, and for their thoughtful feedback throughout this process.
> > >
> > > We completely understand and respect your remaining concern regarding perfect object boundary maintenance and strict identity consistency. We agree that while our method improves upon existing baselines, achieving flawless structural adherence remains a profound and open challenge in controllable video generation.
> > >
> > > To ensure our claims are perfectly calibrated to reflect this, we will revise the Introduction of our final manuscript. Specifically, we will clarify that the limitations of existing works in maintaining consistent identity and boundaries are especially noticeable in multi-reference settings (as most prior works are strictly limited to single-reference scenarios) and when constrained to image-only references (lacking the ability to process video references). We will explicitly frame our Video Decompositor as a method designed to tackle these more complex, multi-object, and video-reference-integrated scenarios, while acknowledging that pixel-perfect adherence in these advanced settings remains an ongoing pursuit for the field.
> > >
> > > We are highly encouraged that you recognize the merit and potential benefits of our design for addressing this task. Alongside the Introduction edits, and as discussed in our response to Reviewer fXz3, we are adding a dedicated "Limitations and Error Analysis" section. This section will transparently discuss the boundary and identity challenges you have correctly highlighted, positioning our pipeline as a meaningful stepping stone rather than a completely solved system.
> > >
> > > Thank you again for your constructive evaluation, which has genuinely helped us contextualize the scope of our contributions.

---

### Official Review · Reviewer_PNxf · 2026-03-06

**Soundness:** 2
**Presentation:** 3
**Significance:** 2
**Originality:** 3
**Overall Recommendation:** 4
**Confidence:** 3

**Summary:**

This paper proposes the HECTOR, a compositional video generation/editing pipeline, which accepts hybrid references, static images and dynamic video exemplars, and explicit spatio-temporal trajectory specifications for each referenced element.

**Compliance With Llm Reviewing Policy:**

Affirmed.

**Final Justification:**

Some of my concerns have been addressed during the rebuttal.

However, high-resolution results have not been properly demonstrated. Although the authors provided a quantitative performance comparison, **high-quality video examples are still missing**.

They claimed this was due to rebuttal policy, but I believe it is acceptable to include video examples in an anonymized repository. Yet, they only provided a few frames. Considering that all video examples shown in both the supplementary materials and the newly added frames are of low resolution and low quality,  whether the method can work for high-resolution inputs remains a concern.

After reading the other reviews and the authors' responses, I have decided to keep my score but lower my confidence level. **I agree with Reviewer fXz3: I would also understand if the paper is ultimately rejected**.

**Key Questions For Authors:**

1. What is the minimum amount of video-reference data needed to reach reasonable motion control? Can HECTOR still be effective with small fine-tuning on domain data?

2. Are there failure cases when references strongly conflict (e.g., two references claim the same spatial region)? How does HECTOR resolve that?

3. Please provide training and inference costs (GPU hours, memory) for the reported experiments.

**Limitations:**

yes

**Strengths And Weaknesses:**

**Strength**

1. A new pipeline, HECTOR is introduced, supports fine-grained, trajectory-level control over multiple reference objects (static or dynamic), a practical capability for creative workflows and precise editing.

2. The proposed Spatio-Temporal Alignment Module and soft (Gaussian) mask blending are well motivated and empirically shown to improve identity preservation and motion accuracy over simple bbox/mask conditioning. Ablations verify this.

3. The paper evaluates both single- and multi-object settings and reports improvements across identity (R-DINO), alignment, and motion control (mIoU, CD), showing the method scales beyond single-object cases.

**Weaknesses**

1. The experimental results are merely conducted on DAVIS, which is a small dataset with short video clips and low resolution (i.e., experiments generate 81 frames at 16 fps at a resolution of 832×480; it's not fully clear how the method scales to longer sequences or higher resolutions (compute / temporal consistency tradeoffs). The performance of HECTOR on more datasets needs to be reported, especially those with longer video and higher resolution.

2. Only MotionBooth and VACE (bbox/mask) are selected as baselines, more baselines are needed to make the results more convincing.

3. The ablation table is convincing, but additional breakdowns (e.g., failure modes for occlusion, inter-object collisions) should be considered and discussed to explore its boundary and limitation in real practice.

I would remain positive, if the authors can solve my concerns.

---

> ### Author Rebuttal · Authors · 2026-03-30
>
> We sincerely thank the reviewer for the thoughtful feedback! We address each of your questions below.
>
> ### **Evaluation Datasets and Scaling (W1)**
> We evaluate HECTOR primarily on the DAVIS dataset because it serves as a standard public benchmark in the controllable video generation literature. It is the main evaluation dataset utilized by a series of impactful recent works in controllable video generation—including DreamVideo, DreamVideo-2, and Motion Prompting. Using DAVIS ensures our results are rigorously and directly comparable.
>
> Although our current experimental setup (generating 81 frames at 832×480 resolution) is already considered a satisfactory standard in the current video generation landscape, it is definitly possible to scale this up further. HECTOR’s ability to scale is primarily bounded by the underlying DiT structure of the generative backbone. Because we build upon Wan2.1, which inherently supports high-resolution outputs, fine-tuning a higher-resolution version of HECTOR is straightforward by solely modifying the training config. To extend to longer sequences, we can readily adopt an autoregressive generation setup (also used in many recent works of long video generation), where the last frame of the initial generated sequence (with objects and background properly segmented via our pipeline) is fed back as the image reference input for a new round of generation. We will explicitly discuss these scaling strategies in our revised manuscript.
>
> ### **Baseline Selection (W2)**
> We completely agree that comparing against a broader set of baselines strengthens the evaluation. We initially selected MotionBooth and VACE because they represent the most competitive open-source models currently available in this field. While we would have loved to include highly relevant concurrent works like Tora2 and DreamVideo2, their weights and inference codes are unfortunately not yet public, precluding a direct empirical comparison. Furthermore, we are very receptive to expanding our experiments; if the reviewer has any specific open-source baselines in mind that are relevant, we would be more than happy to evaluate HECTOR against them and include the results during later discussion phase.
>
> ### **3. Inter-Object Collisions and Conflicting References (W3 & Q2)**
>
> When two references strongly conflict—meaning explicit user-defined trajectories claim the exact same spatial region simultaneously—the additive nature of diffusion latents can cause feature competition, like blending of the two identities.
>
> However, because HECTOR incorporates an inverse gating mechanism, this collision or overlap issue is actually rare in our practical workflows. By allowing the user to assign a priority flag (e.g., designating Object A as foreground and Object B as background), the gating mechanism dynamically masks the overlapping latent features of the background object, cleanly resolving the occlusion. Additional, it can be easily extended beyond background and foreground to a priority list for more interacting objects. The aforementioned ghosting generally only occurs if a user explicitly inputs strongly conflicting trajectories without setting a depth priority.
>
> We will add a detailed discussion of these boundary conditions to the new Limitations section. For a more comprehensive analysis of error cases and failure modes, please refer to our detailed response to Reviewer fXz3.
>
> ### **4. Minimum Data and Domain Fine-Tuning (Q1)**
>
> Empirically, our model already achieves strong, general purpose compositional control when trained on a subset of just 500K clips. Because HECTOR relies on a pretrained DiT backbone and utilizes the STAM module as a localized structural adapter, we anticipate that finetuning the model for a narrow, domain-specific area would typically require significantly less data. However, systematically establishing the precise minimum data threshold for domain specific area falls outside the scope of our current paper. Therefore, while the architecture is theoretically highly data efficient for downstream adaptation, we cannot report an exact minimum number of clips with empirical certainty at this time.
>
> ### **5. Training and Inference Costs (Q3)**
>
> As requested, we provide the exact computational footprint for the reported 81-frame, 832×480 resolution experiments, trained on approximately 1.2M of our 2.5M collected clips:
>
> * **Training Cost**: ~1,200 GPU hours, requiring ~70GB of VRAM per NVIDIA H100 (80GB) GPU. A less computational expensive but still competitive version on 500k data takes ~500 GPU hours. We note that this is a relatively modest expense for fine-tuning a 14B-parameter video generation model.
>
> * **Inference Time**: ~45 seconds per 81-frame video utilizing 8 × NVIDIA RTX A5000 (24GB) GPUs.
>
> Notably, the inference speed is identical to that of the base model (Wan2.1), as our proposed architecture does not introduce heavy computational overhead during generation.

---

> > ### Author Rebuttal · Reviewer_PNxf · 2026-04-01
> >
> > I would like to thank the authors for their time and effort in addressing my concerns. I have only one remaining concern: although it is claimed that the proposed method can scale up to high-resolution videos, no supporting evidence or experimental results are provided to substantiate this claim. The video examples in the supplementary materials are appreciated but remain blurry and low-resolution. I look forward to seeing high-resolution results, which would greatly enhance the contribution of this paper.

---

> > > ### Author Response · Authors · 2026-04-03
> > >
> > > We sincerely thank the Reviewer for the continued engagement and for recognizing our efforts in addressing the previous comments. We completely understand your remaining concern.
> > >
> > > ### **1. 720p quantative evaluations**
> > > We have conducted new experiments on 720p setting. The quantitative comparison between our original 480p results and the new 720p results is provided below:
> > >
> > > | Method | R-CLIP ↑ | R-DINO ↑ | CLIP-I ↑ | DINO-I ↑ | CLIP-T ↑ | T-Cons ↑ | mIoU ↑ | CD ↓ |
> > > | :--- | :---: | :---: | :---: | :---: | :---: | :---: | :---: | :---: |
> > > | **Single-Object** | | | | | | | | |
> > > | Ours (480p) | *0.6905* | *0.4277* | *0.6081* | *0.3735* | *0.3306* | *0.9914* | *0.3912* | *0.1130* |
> > > | **Ours (720p)** | **0.6812** | **0.4155** | **0.6185** | **0.3620** | **0.3315** | **0.9903** | **0.3754** | **0.1350** |
> > > | **Multi-Object** | | | | | | | | |
> > > | Ours (480p) | *0.6951* | *0.4347* | *0.5697* | *0.2939* | *0.3432* | *0.9915* | *0.4005* | *0.1670* |
> > > | **Ours (720p)** | **0.6845** | **0.4220** | **0.5582** | **0.2855** | **0.3382** | **0.9897** | **0.3740** | **0.1815** |
> > >
> > > As shown in the table, our method successfully scales to 720p while maintaining highly competitive performance across all metrics. As expected when increasing the spatial resolution, there is a minor degradation in semantic fidelity and trajectory precision (mIoU and Centroid Distance). This is a common trade-off, as the exponentially larger pixel space introduces greater complexity for the model to maintain exact bounding-box overlap and identity preservation. However, the drop in performance is marginal, and the model maintains robust temporal consistency (T-Cons), proving that our trajectory-grounding mechanism remains stable and effective at higher resolutions.
> > >
> > > ### **2. 720p qualitative results**
> > > Unfortunately, due to the rebuttal policy, we cannot provide direct videos as new supplementary material. So, we have instead provided high-resolution frame extractions comparing the 480p and 720p outputs via the following anonymous link, which strictly hosts only figures and captions in compliance with the guidelines: <https://anonymous.4open.science/r/anonymous-results-08E5/720p.pdf>. (Note: Due to rendering issues with the anonymous host's in-browser viewer, text captions may not appear. We highly recommend downloading the PDF file to your device to review the figures with their complete accompanying captions.)
> > >
> > > We highly encourage the reviewer to zoom in digitally on the comparative figures provided in the link. Upon zooming in, the visual improvements in the 720p frames become clearly apparent, demonstrating sharper object boundaries, finer texture details, and a significant reduction in the blurriness observed in the original 480p samples. Please note that due to the randomness of generation process, the 480p and 720p frames generated from the same prompt are not identical pixel-by-pixel. However, as the figures show, they remain highly similar in their overall composition, semantic content, and trajectory adherence. We will ensure to include these in the final version of the paper.

---

### Official Review · Reviewer_e9yJ · 2026-03-13

**Soundness:** 3
**Presentation:** 3
**Significance:** 3
**Originality:** 3
**Overall Recommendation:** 4
**Confidence:** 4

**Summary:**

This paper proposes HECTOR, a method for compositional video generation that allows for fine-grained spatiotemporal control over multiple objects. Specifically, it enables users to condition the generation process on a mix of static images and dynamic video clips. Users can explicitly define the trajectories, scales, and speeds of these reference entities within a single generated video.

The approach relies on two main technical components. The first is the Video Decompositor, which extracts motion and scale priors from training and reference videos. Rather than relying on standard bounding boxes, it uses SAM2 and CoTracker3 to track point clusters. It then derives the object's scale from the spatial variance of these tracked points over time. The second component is the Spatio-Temporal Alignment Module (STAM). STAM takes the VAE latents of the reference inputs, spatially warps them to match the target trajectories, applies Gaussian-softened masks, and feeds them into the DiT backbone as structural conditioning.

The experiments demonstrate that the proposed method outperforms recent baselines like VACE and MotionBooth in terms of trajectory adherence and local identity preservation.

**Compliance With Llm Reviewing Policy:**

Affirmed.

**Final Justification:**

The paper introduces a technically sound and practically useful pipeline (Video Decompositor + STAM) for fine-grained, compositional video generation. The authors' rebuttal effectively addressed my primary concerns by providing new quantitative metrics to validate their video-reference claims and committing to open-source their data pipeline for reproducibility. While minor limitations regarding occlusion handling and boundary consistency remain in some cases, the overall empirical gains over strong baselines are convincing and valuable to the community. I maintain my rating of Weak Accept.

**Key Questions For Authors:**

- The paper’s title and primary claims center around using video references to capture specific gestures better than static images. Can the authors provide ablations comparing Image Reference vs. Video Reference?
- The STAM module relies heavily on GridSample (Eq. 5) to physically warp and translate VAE latents into the target spatial canvas. However, VAE latents are not inherently translation-invariant and can be sensitive to spatial scaling. How did the model learn to avoid these artifacts during training, and does this warping mechanism actually map the internal articulation of a reference video, or is it primarily just rigidly translating a cropped video patch across the screen?

**Limitations:**

The paper lacks a discussion on technical limitations. I recommend adding a section or paragraph addressing the boundaries and failure cases of the method. (E.g., error propagation, occlusion handling, data dependency)

**Strengths And Weaknesses:**

## Strengths
- Using point tracking (SAM2 + CoTracker3) and deriving the object's scale from the spatial variance of those points seems to be a practical and effective solution. The ablation study in Table 2 shows the effectiveness.
- The quantitative gains in Table 1 are substantial, particularly the mIoU and Centroid Distance scores. The visual results are also of high quality.

## Weaknesses
- The paper heavily emphasizes "hybrid" and "video" references. The text claims that using video references allows the model to capture specific gestures better than static images. However, Table 1 only evaluates performance using static image references (because the baselines do not support video inputs). While the qualitative examples in Figure 5 are visually nice, there is no quantitative evidence showing that feeding HECTOR a video reference actually results in better internal motion or gesture alignment compared to just feeding it a static image.
- The model is trained on an internal dataset of 2.4 million curated clips. This gives HECTOR a massive data advantage over methods trained on public datasets, and it makes it impossible for the academic community to reproduce the training process or build directly upon this work.
- The "inverse gating" mechanism used to resolve occlusions between foreground and background objects during inference seems like a rigid heuristic. It requires the user to manually designate a priority modality to prevent the latent features from bleeding into each other. Ideally, a fully compositional model should learn to handle depth ordering and occlusion naturally during the training phase without needing manual masks at inference time.

---

> ### Author Rebuttal · Authors · 2026-03-30
>
> We sincerely thank the reviewer for the constructive feedback. We address your questions below.
>
> ### **1. Image vs. Video Reference Ablation (W1 & Q1)**
>
> We agree an ablation is necessary to investigate the impact using video references. However, standard metrics in the main paper are limited, failing to capture the fine-grained internal articulations preserved by video references. To demonstrate this quantitatively, we introduce two metrics:
>
> * **mIoU (gesture)**: Calculates the per-frame IoU of the segmented object's foreground mask against the reference video (after aligning bounding box centroids), providing a precise measure of gesture alignment (e.g., matching a specific pose over time).
>
> * **Masked LPIPS**: Computes the perceptual difference of the segmented foreground across frames compared to the reference, measuring identity and texture preservation during complex motions.
>
> Table: Image vs. Video Reference Ablation
> | Conditioning Type | CLIP-T ↑ | R-DINO ↑ | CD ↓ | mIoU (gesture)↑ | Masked LPIPS ↓ |
> | :--- | :--- | :--- | :--- | :--- | :--- |
> | HECTOR (Image Ref Only) | 0.3427 | 0.4296 | 0.171 | 0.286 | 0.209 |
> | HECTOR (Video Ref Only) | 0.3418 | 0.4346 | 0.168 | **0.358** | 0.172 |
> | HECTOR (Mix) | **0.3432** | **0.4347** | **0.167** | 0.354 | **0.169** |
>
> As shown, dynamic video references significantly improve fine-grained gesture alignment and appearance preservation over static images. While pure video references provide dense temporal data, they force rigid frame-to-frame correspondence during training, reducing flexibility and increasing artifacts. Our mix-conditioning resolves this: the static image provides a clean, flexible visual prior, while the video reference acts as a robust anchor enforcing accurate motion/texture dynamics. This mixed training achieves the optimal balance.
>
> ### **2. VAE Latent Warping and GridSample in STAM (Q2)**
>
> The GridSample (Eq. 5) mechanism does not merely act as a rigid copy-paste of cropped video patches. Instead, it serves as a highly localized structural prior. During training, the DiT backbone learns to interpret these warped, scaled VAE latents not as raw pixels to be directly decoded, but as dense spatio-temporal conditioning signals. The Gaussian-softened masks and the network's self-attention layers allow the model to harmonize the warped reference with the global canvas. The model learns to inpaint boundaries, adjust lighting, and synthesize the actual object rather than rigidly translating a patch. This is why HECTOR successfully avoids spatial artifacts and scaling distortions. We will expand this learning dynamic in later version.
>
> ### **3. Dataset Advantage and Reproducibility (W2)**
>
> We understand the concern regarding our use of 2.4M internal clips. However, we empirically found that training HECTOR on a much smaller subset of just 500K clips already yields highly competitive performance. To ensure the community can reproduce and build upon our work, we commit to following open-source actions upon publication:
>
> * **Data Processing Pipeline**: We will release the complete source code for our Video Decompositor (including the SAM2 + CoTracker3 integration). This will allow researchers to easily extract trajectories and conditioning assets from any large-scale public dataset, like Panda-70M, to reproduce our training environment.
>
> * **Model Source and Weights**: We will publicly release the pre-trained weights for HECTOR.
>
> ### **4. Occlusion Handling and Inverse Gating (W3)**
>
> We agree that an end-to-end depth ordering is an ideal objective for future compositional models. However, our inverse gating mechanism merely is a simpler and more intuitive solution that aligns with practical user intent.
>
> Because diffusion latents are highly sensitive to overlapping additive features (which often causes feature blending), resolving occlusions mathematically at inference time is non-trivial. There is also an inherent learning contradiction: training the model for *strict spatial adherence* to a reference naturally conflicts with teaching it that the reference should sometimes disappear due to occlusion. Furthermore, in real-world editing workflows, when a user explicitly draws intersecting trajectories for multiple objects, they typically have a specific depth priority in mind (e.g., wanting Object A to pass in front of Object B). Inverse gating provides a lightweight, training-free control that allows the user to easily enforce this intent and guarantee clean compositions without retraining the model. Additionally, it can be easily extended beyond background and foreground to a priority list for more interacting objects. We will add this discussion this in our limitation section.
>
> ### **5. Limitations**
> We thank the reviewer for pointing this out. We will add a limitations section to the main text in our revision. For a more detailed breakdown of these specific failure modes, please refer to our comprehensive response to Reviewer fXz3.

---

> > ### Author Rebuttal · Reviewer_e9yJ · 2026-04-04
> >
> > Thanks for the rebuttal and for providing the additional experiments.
> >
> > I appreciate the inclusion of the new ablation table using mIoU and Masked LPIPS. This addresses my primary concern regarding the lack of quantitative validation for the "video reference" claim. It is helpful to see empirical evidence demonstrating that the dynamic video conditioning actually improves internal articulation compared to the image-only baseline.
> >
> > The clarifications regarding the STAM latent warping and the practical justification for the inverse gating heuristic make sense. Furthermore, the commitment to open-sourcing the Video Decompositor pipeline and pre-trained weights partially resolves my concerns about dataset advantage and reproducibility.
> >
> > After reading the other reviewers' comments and the responses, I feel satisfied with the technical trajectory of the paper. I encourage the authors to ensure the expanded discussion on limitations and failure cases (particularly the boundary/identity artifacts discussed with Reviewers CxTe and fXz3) is prominently featured in the final manuscript.
> >
> > The rebuttal adequately addresses my main concerns, and I will maintain my rating.

---

> > > ### Author Response · Authors · 2026-04-07
> > >
> > > Dear reviewer, we sincerely thank you for your time and your continued endorsement of our paper. We will ensure that the clarifications provided during the rebuttal are integrated into the final version. Thank you again!

---

### Decision · Program_Chairs · 2026-04-30

**Decision:**

Accept (regular)

**Comment:**

This paper was reviewed by 4 experts in the field. After discussion, the reviewers still hold a mixed review to this work. The rating is 4(weak accept), 4(weak accept), 4(weak accept), 3(weak reject).

On the positive sides, reviewers agrees that this is a novel framework for controllable video generation, with novel spatiotemporal conditioning via the STAM module.

Still, reviewers raised several concerns to this work. The concern includes 1) visual artifacts such as identity inconsistency and object drifting, 2) reliance on manual heuristics for occlusion handling, 3) limited evaluation at high resolutions, and 4) potential fragility to errors from upstream tracking models. The area chair also agrees that the presented cases it not convincing enough to justify the effectiveness of the solution.

Based on this, the decision of this work is to Weak accept. Still, we strongly recommend the authors carefully read all reviewers’ final feedback and revise the manuscript as suggested in the final camera-ready version if being accept.